

# Flood algorithm: a novel metaheuristic algorithm for optimization problems

Ramazan Ozkan[1,2] and Ruya Samli[2]

[1] Department of Computer Engineering, National Defence University, Istanbul, Turkey
[2] Department of Computer Engineering, Istanbul University-Cerrahpasa, Istanbul, Turkey

## ABSTRACT

Metaheuristic algorithms are an important area of research that provides significant advances in solving complex optimization problems within acceptable time periods. Since the performances of these algorithms vary for different types of problems, many studies have been and need to be done to propose different metaheuristic algorithms. In this article, a new metaheuristic algorithm called flood algorithm (FA) is proposed for optimization problems. It is inspired by the flow of flood water on the earth's surface. The proposed algorithm is tested both on benchmark functions and on a real-world problem of preparing an exam seating plan, and the results are compared with different metaheuristic algorithms. The comparison results show that the proposed algorithm has competitive performance with other metaheuristic algorithms used in the comparison in terms of solution accuracy and time.

## TERMS AND CONDITIONS

The terms and definitions used in this study are given in Table 1.

## INTRODUCTION

Optimization can be defined as the attempt to achieve the best result by making choices on the inputs of an objective function within the framework of certain criteria. It is a frequently encountered problem in almost every field of science and real-world problems. Solving complex optimization problems (*e.g.*, the NP-complete problems) with conventional gradient-based methods is a time-consuming process (*Chen, Cai & Wang, 2018*). The complexity of the problem makes it impossible to search for all possible solutions or combinations in an acceptable time frame (*Yang, 2010b*).

Metaheuristic algorithms are an impressive area of research that has made significant advances in solving challenging optimization problems (*Dokeroglu et al., 2019*). They use random operators, trial-and-error processes and random scanning of the problem solving space to generate efficient solutions to optimization problems (*Dehghani et al., 2023*). The optimization process in meta-heuristic algorithms starts by generating a certain number of random feasible solutions in the problem space. In an iterative process, candidate solutions are updated and improved according to the algorithm instructions. After the algorithm is fully implemented, the best solution among the candidate solutions is presented as the solution to the problem (*Dehghani et al., 2020*).

Corresponding authors
Ramazan Ozkan,
rozkan@hho.msu.edu.tr
Ruya Samli, ruyasamli@iuc.edu.tr

**Table 1 Terms and definitions.**

| Term | Definition |
| --- | --- |
| Metaheuristic algorithm | A high-level procedure or heuristic based on a heuristic designed to provide a sufficiently good solution to optimization problems that are difficult to solve with traditional gradient-based methods. |
| Metaheuristic parameter | Algorithm-specific inputs to reflect the nature of the phenomena that metaheuristics are inspired by. |
| Decision variable | Variables whose values can be changed over the set of feasible alternatives to increase or decrease the value of the objective function in an optimization problem. |
| Solution | Each set of decision variables formed by the values of the decision variables on the set of feasible alternatives. |
| Neighbour solution | The set of decision variables formed by changing the value of any variable in the decision variable set of an existing solution. |
| Fitness value | The value of the objective function corresponding to any set of values of the decision variables. |
| Operator | The manipulations that metaheuristic algorithms make on the solutions that emerge during the search process in order to efficiently scan the search space and converge on the best solutions. |
| Local optimization | Finding the optimal solution for a specific region of the search space. |
| Global optimization | Finding the global minimum or maximum of a function or set of functions on a given set. |
| Convergence | The tendency of the solutions obtained in each iteration to get closer to the desired solution. |
| Exploration | Exploring the global solution space by generating different solutions. |
| Exploitation | The search of the neighbourhood of a promising region. |
| Complexity | The amount of resources (such as time or memory) required to solve a problem or perform a task. |
| Population-based metaheuristic | The metaheuristics that use a group of points, called a population, to explore the search space. |
| Trajectory-based metaheuristic | The metaheuristics that start the search process from a single starting point and move through the search space with a single solution. |
| Evolutionary algorithms | The metaheuristics follow the natural evolutionary process found in nature. |
| Swarm intelligence algorithms | The metaheuristics that mimic the social behavior of groups of insects or animals. |
| Benchmark function | The functions used to test the performance of any optimization algorithm. |
| Wilcoxon Rank-Sum Test | A non-parametric statistical test used to compare two independent measurements set to assess whether their population mean ranks differ. |
| Rank | The value given to each sample to determine if one group has higher or lower scores than another group. |
| Exam seating problem | The process of allocating seats efficiently with minimal resource usage under certain conditions in an exam. |
| Exam session | A set of exams conducted at the same time. |
| Traceability matrix | A matrix representing all possible classrooms and seat locations and providing access to seat location neighbours through index information in the solution vector. |

Simple concepts, easy implementation, no need for a derivation process, efficiency in high-dimensional problems, efficiency in non-linear and non-convex environments are some of the advantages that lead to the popularity and widespread use of metaheuristic algorithms (*Cavazzuti, 2013*). These advantages and their ability to solve problems in a wide variety of domains without knowing the details and definitions of the problems, and to provide near-optimal solutions, gives them an advantage over traditional techniques (*Rajpurohit et al., 2017*). Therefore, in recent decades, a number of metaheuristic algorithms have been proposed and successfully applied to solve complex optimization problems in various scientific domains and real-world problems. According to the number of related studies, well-known and most used algorithms are: Genetic Algorithm (GA) (*Holland, 1992*), Simulated Annealing (SA) (*Kirkpatrick, Gelatt & Vecchi, 1983*), and Particle Swarm Optimization (PSO) (*Eberhart & Kennedy, 1995*). There are numerous

studies in the literature where these algorithms, improved versions, or hybrid versions of these algorithms are proposed or applied to a problem.

Although metaheuristic algorithms show strong optimization ability in solving nonlinear global optimization problems, some of them fall into local optima when faced with different optimization problems (*Yang, 2010b*). Moreover, a metaheuristic may perform well on one set of optimization problems but poorly on another (*Wolpert & Macready, 1997*).

These fundamental gaps encourage the discovery and development of new metaheuristic algorithms with satisfactory performance. For this reason, there is an increasing amount of work in the literature to propose new metaheuristic algorithms. In line with these motivations, this article proposes a new metaheuristic algorithm called the flood algorithm.

The contributions of this study can be expressed as follows:

- A new optimization algorithm called Flood Algorithm (FA) is developed to model the flow of flood water.
- Fifteen standard benchmark functions, eight of them from the CEC2022 test suite, and three engineering design problems were used to evaluate the performance of FA in solving optimization problems.
- The performance of FA in real-world applications is tested by solving an exam seating problem, which is a permutation problem and consists of eleven sessions.
- The performance of FA is validated against the three well-known metaheuristic algorithms.

The main advantage of the proposed FA approach for global optimization problems is that it has a single basic parameter and thus can be easily adapted to different optimization problems. The second advantage of FA is its highly effective efficiency in dealing with high-dimensional optimization problems. The third advantage of the proposed method is its strong performance in handling real-world optimization applications.

The rest of the article is organized as follows: "Literature Review" section presents a literature review about metaheuristic algorithms. "Flood Algorithm" section describes the proposed flood algorithm. "Flood Algorithm Validation" section presents the application and results of the proposed algorithm and three other algorithms on benchmark functions and a real-world problem and compares the results, and "Conclusion" section concludes this article and provides suggestions for future work.

## LITERATURE REVIEW

Metaheuristic algorithms are inspired by nature and mimic biological and physical processes such as the behavior of animals, insects, birds, living things, physical laws, biological sciences, human activities, rules of the games, and any other evolution-based process to solve optimization problems (*Tanhaeean, Tavakkoli-Moghaddam & Akbari, 2022*; *Dehghani et al., 2023*). For example, the GA (*Holland, 1992*) mimics evolutionary processes in nature, PSO (*Eberhart & Kennedy, 1995*) mimics swarm movement of birds or

fish while the SA (*Kirkpatrick, Gelatt & Vecchi, 1983*) mimics physical metal annealing processes. Based on the development of various metaheuristic algorithms in the last decades, Metaheuristic algorithms can be divided into three main categories: Evolutionary Algorithms (EA), which follow the natural evolutionary process found in nature. Swarm Intelligence Algorithms (SI), which include swarm-based techniques that mimic the social behavior of groups of insects or animals. And other metaheuristic algorithms that mimic principles of physics, chemistry, gaming and human behavior (*Azizi, Talatahari & Gandomi, 2023*; *Hashim et al., 2022*; *Ayyarao et al., 2022*).

EA are based on the mechanism of natural selection. They mimic the way natural evolution and genetic mechanisms works. GA and Differential Evolution (*Storn & Price, 1997*) are among the most widely used evolutionary algorithms designed based on the modeling of the reproductive process, natural selection, Darwin's theory of evolution, and the use of random operators of selection, crossover, and mutation (*Dehghani et al., 2023*). Some other algorithms in this group are: Genetic Programming (*Koza, 1992*), Evolution Strategy (*Bäck, 1995*), Covariance Matrix Adaptation Evolutionary Strategy (*Hansen & Ostermeier, 2001*), Wild Horse Optimizer (*Naruei & Keynia, 2022*) and, Biogeography-Based Optimization (*Simon, 2008*).

SI algorithms have been developed inspired by natural swarming phenomena, the collective and self-organizing behavior of birds, fishes and, other living things in nature. Some of the known meta-heuristic algorithms are: PSO (*Eberhart & Kennedy, 1995*), Ant Colony Optimization (*Dorigo, Maniezzo & Colorni, 1996*), Firefly Algorithm (*Yang, 2010a*), Gray Wolf Optimization (*Mirjalili, Mirjalili & Lewis, 2014*), Whale Optimization Algorithm (*Mirjalili & Lewis, 2016*), White Shark Optimizer (*Braik et al., 2022*), Harris Hawks Optimizer (*Heidari et al., 2019*), War Strategy Optimization Algorithm (*Ayyarao et al., 2022*), Shrimp and Goby Association Search Algorithm (*Sang-To et al., 2023*), Orchard Algorithm (*Kaveh, Mesgari & Saeidian, 2023*), Gannet Optimization Algorithm (*Pan et al., 2022*), Dung Beetle Optimizer (*Xue & Shen, 2023*) and Honey Badger Algorithm (*Hashim et al., 2022*).

The third group algorithms are developed based on mathematical modeling of various events, concepts, laws, and forces in physics, chemistry, games, and human behavior. Some of the well-known metaheuristic algorithms are: SA (*Kirkpatrick, Gelatt & Vecchi, 1983*), Gravitational Search Algorithm (*Rashedi, Nezamabadi-Pour & Saryazdi, 2009*), Group Teaching Optimization Algorithm (*Zhang & Jin, 2020*), Henry Gas Solubility Optimization (*Hashim et al., 2019*), Fireworks Algorithm (*Tan & Zhu, 2010*), Chaos Game Optimization (*Talatahari & Azizi, 2021*), Harmony Search Algorithm (*Yang, 2009*), Hunger Games Search (*Yang et al., 2021*), Boxing Match Algorithm (*Tanhaeean, Tavakkoli-Moghaddam & Akbari, 2022*), Atomic Orbital Search (*Azizi, 2021*).

From another perspective, metaheuristics can be classified into two groups: trajectory-based metaheuristics and population-based metaheuristics. The main difference between these two types of methods is based on the number of solutions used at each step of the algorithm. Trajectory-based algorithms use a single agent or solution that moves through the design space or search space. Population-based algorithms, on the other hand, use a large number of agents or solutions at each search step. Population-based algorithms are

much more popular in the literature than trajectory-based algorithms. For example, EA and SI are all population-based algorithms. The most common examples of trajectory-based algorithms are: SA (*Kirkpatrick, Gelatt & Vecchi, 1983*), Tabu Search (*Glover, 1989*), Iterated Tabu Search (*Misevicius, Lenkevicius & Rubliauskas, 2006*), Guided Local Search (*Davenport et al., 1994*) and Variable neighbourhood Search (*Mladenović & Hansen, 1997*).

# FLOOD ALGORITHM

In this section, a new metaheuristic algorithm called the flood algorithm is discussed in detail. The FA is a metaheuristic algorithm for optimization problems that can be used to approach global optimization in a large search space within an acceptable time frame. Metaheuristic algorithms are divided into two groups according to their search mechanisms: trajectory-based algorithms, which work on a single solution at each iteration, and population-based algorithms, which work on a collection of solutions. The FA is a trajectory-based algorithm. It starts with a random solution and searches for a global optimum by manipulating that solution at each iteration.

## Inspiration

The design of the FA was inspired by the movement of floodwaters on the earth's surface. It mimics the gravity-driven movement of flood waters toward lower elevations and overcoming obstacles with the kinetic energy they gain from this movement. The movement of flood waters follows two basic rules. The first rule is the natural flow of water, which means that water always flows toward lower elevations under the influence of gravity. The second rule is that when water encounters an obstacle, it accumulates at that point and continues to flow by overcoming the obstacle at its lowest point.

## Mathematical model

This section describes the mathematical formulation of the proposed FA. Since the FA includes both exploration and exploitation processes, it can be considered a global optimization algorithm. The pseudo-code of the FA including the initialization, search and evaluation, and the termination steps is presented in Algorithm 1, and the flowchart is presented in Fig. 1. Mathematically, the steps of the proposed algorithm are detailed as follows.

### *Initialization*

The FA is a trajectory-based algorithm starting from a single starting point $X_{init}$ as defined in Eq. (1) and with a starting speed $V_{init}$ of 1.

$$X_{init} : [x_1, \ldots, x_i, \ldots, x_n], \quad i = 1, 2, \ldots, n,$$
$$n \text{ is the number of decision variables} \tag{1}$$

This point represents a candidate solution to the problem for which the values of the decision variables are randomly determined using Eq. (2) and represents the location where the flood starts.

---

**Algorithm 1  Pseudo-code of the flood algorithm.**

$m \leftarrow$ maximum number of neighbours to ˘in every iteration

$v_{init} \leftarrow 1, v_{min} \leftarrow 0.1, v \leftarrow v_{init}$

$s \leftarrow s_0$                                                  ▷ initial state

$s_{best} \leftarrow s_0$

**while** $v > v_{min}$ **do**

    **for** $k = 1$ to $m$ **do**

        $s_{new} \leftarrow$ get a random neighbour of s

        **if** $f(s_{new}) < f(s)$ **then**                   ▷ fitness value comparison

            $v \leftarrow v * (f(s)/f(s_{new}))$            ▷ speed update

            $s \leftarrow s_{new}$

            **if** $f(s) < f(s_{best})$ **then**

                $s_{best} \leftarrow s$

        **break**

        $neighbours[k] \leftarrow s_{new}$

    $v \leftarrow v * (f(s)/f(\min(neighbours)))$        ▷ speed update

    $s \leftarrow \min(neighbours)$

---

$$X_{init} : x_i = lb_i + r \times (ub_i - lb_i) \tag{2}$$

where $x_i$ is the value of the $i$th decision variable, $r$ is a random real number in the range [0, 1], and $lb_i$ and $ub_i$ are the lower and upper bounds of the $i$th decision variable, respectively. After evaluating the fitness of the starting point, the FA runs in iterations until the termination condition is met. The starting point $X_{init}$ is used as the current solution $S_{cur}$ and starting speed $X_{init}$ is used as the current speed of the first iteration with Eq. (3).

$$S_{cur} = X_{init} \\ V_{cur} = V_{init} \tag{3}$$

### Finding a neighbour solution

The search process of the FA involves finding a certain number of neighbours of the current solution and checking their fitness values in each iteration. A neighbour solution is generated by randomly updating the value of one of the variables of the current solution within the bounds of that variable. Suppose the optimization problem has $n$ variables. The current solution is $X = x_1, \ldots, x_k, \ldots, x_n$ and the variable $x_k$ (whose lower and upper bounds are $lb_k$ and $ub_k$) is selected for variation. The new value of the selected variable is randomly determined by using Eq. (4) and the neighbour solution is obtained as seen in Eq. (5).

$$x'_k = lb_k + r \times (ub_k - lb_k) \tag{4}$$

$$S_{new} : [x_1, \ldots, x'_k, \ldots, x_n] \tag{5}$$

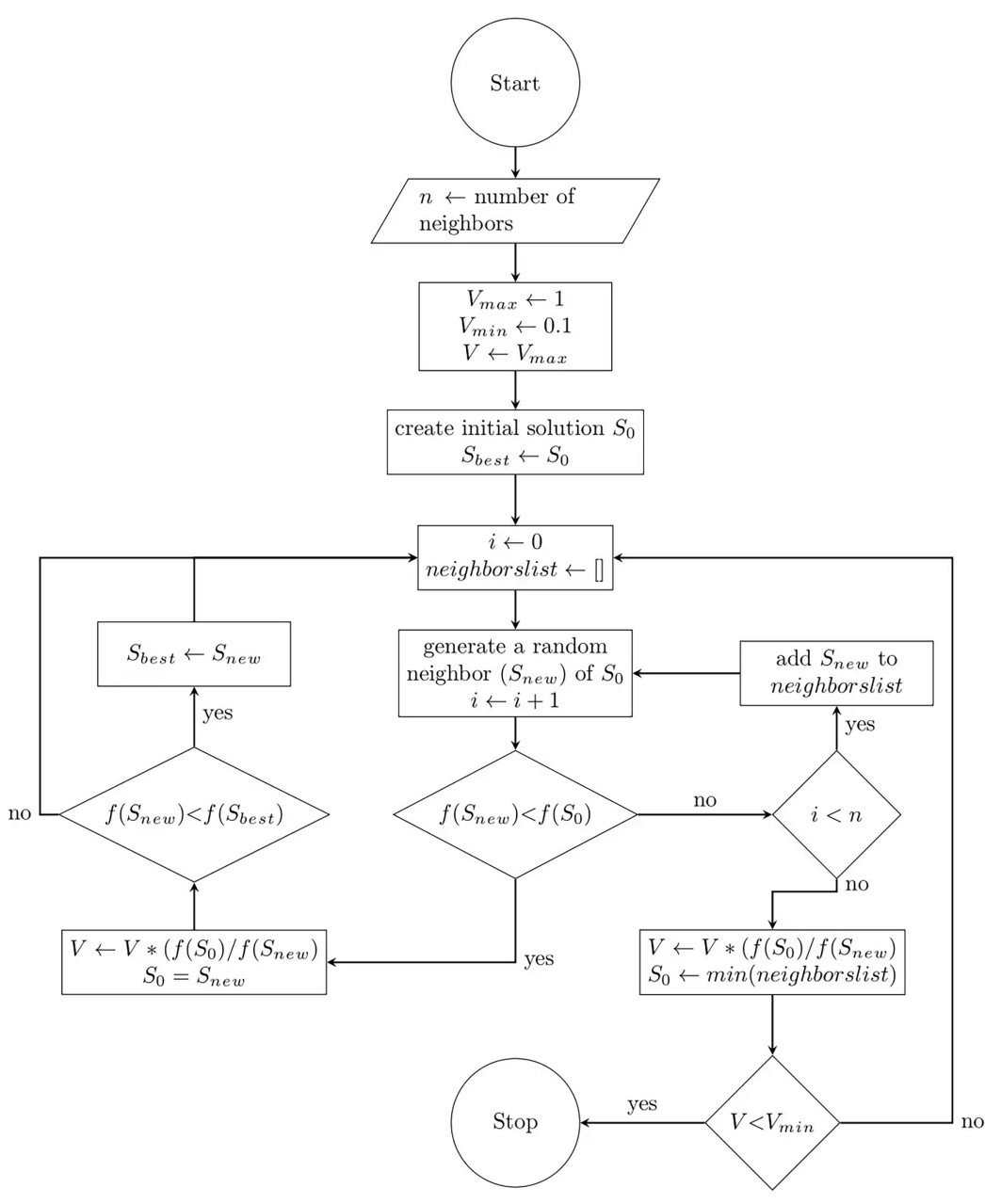

**Figure 1** **Flow chart of the flood algorithm.**           

### Search process

The search mechanism of the FA is based on the two rules of flood movement. It is designed to search the solution space by scanning a predetermined number of neighbour points as defined in Eqs. (4) and (5) of the current solution in each iteration. The current solution $S_{cur}$ is used as the reference point in each iteration and updated at the end of each iteration. When a neighbour with a better (smaller) fitness value than the current solution is found in an iteration as seen in Fig. 2A, it stops scanning the neighbours and continues

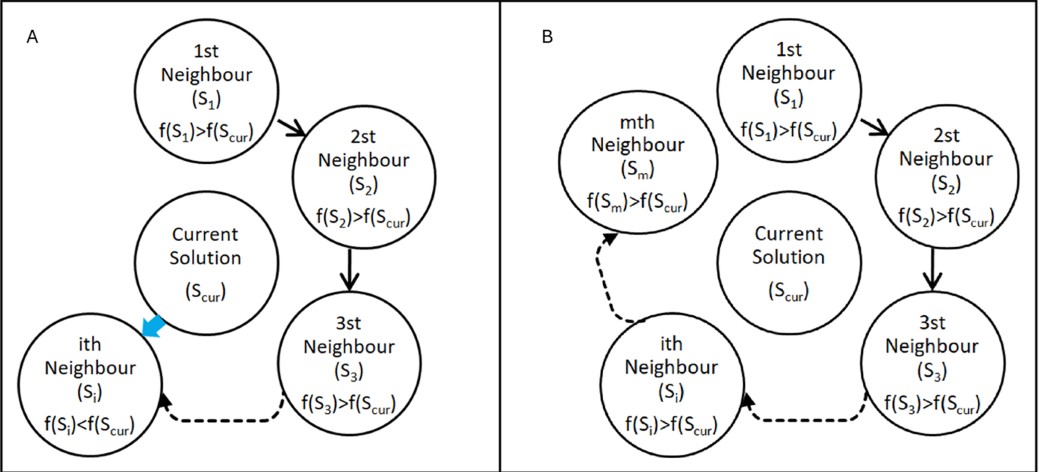

**Figure 2 Search cases of the Flood algorithm.** (A) ith neighbour has a better fitness value than the current solution (B) none of the neighbours has a better fitness value than the current solution.

searching the solution space over that point (modeling the flow of water towards lower elevations) in the next iteration as seen in Eq. (6).

$$S_{cur} = S_i, f(\mathbf{S_i}) < f(\mathbf{S_{cur}})$$

(6)

$S_i$ is the first neighbour found with a better fitness value than the current solution

If a given number of neighbours are scanned and no neighbour with a better fitness value is found as seen in Fig. 2B, in the next iteration it continues to search the solution space over the point with the best fitness value among the neighbours (modeling the flow of water by overflowing from the weakest point of the obstacles it encounters) as defined in Eq. (7).

$$S_{cur} = S_i, S_i \in \{S_1 \ldots S_m\} \quad \wedge \quad f(\mathbf{S_i}) : \min\{f(\mathbf{S_1}) \ldots f(\mathbf{S_m})\}$$

(7)

$m$ is the maximum number of neighbours

In both cases, the speed is updated in proportion to the ratio of the fitness values of the current solution and the new solution which is selected as the current solution for the next iteration as seen in Eq. (8).

$$V_{cur}^{t+1} = V_{cur}^t \times \frac{f(\mathbf{S_{cur}})}{f(\mathbf{S_{new}})},$$

(8)

$t$ refers the current iteration

The speed increases when the new solution has a better fitness value and decreases when the new solution has a worse fitness value. Also; as part of the search mechanism, the FA

maintains the best solution ($S_{best}$) found so far by repeatedly updating it during the search process.

*Termination*

The termination process of FA mimics the stopping of a flood by decreasing its speed in areas where the slope decreases or there is a reverse slope. The FA starts the search from a randomly chosen starting point with an initial speed ($v_{init}$) of 1. It maintains the speed $v$ by repeatedly updating it at each iteration based on the ratio of the fitness value of the current solution $f(S_{cur})$ and the fitness value of the solution selected for the next iteration $f(S_new)$ as defined in Eq. (8). If the fitness value of the solution selected for the next iteration is better than the current solution, the speed is increased, and if it is worse, the speed is decreased. The algorithm stops when the speed falls below the minimum speed ($v_{min}$) of 0.1.

The most commonly used stopping conditions for metaheuristics in the literature are the maximum number of iterations and the maximum number of consecutive iterations without improving the current solution (*Corominas, 2023*). These conditions can be easily applied to the FA as well as to many other metaheuristics. In particular, the maximum number of iterations is a fair way to compare metaheuristics. Therefore, the maximum number of iterations is used as a stopping condition in the FA to compare it with other algorithms.

## Parameters

Metaheuristic algorithms have algorithm-specific parameters to reflect the nature of the phenomena they are inspired by. For example, the GA is a metaheuristic algorithm inspired by the process of natural evolution. It has parameters such as the number of individuals in the population, the crossover rate, the mutation rate, the number of generations, and so on to reflect the evolutionary process. In metaheuristic algorithms, the solution is sensitive to the parameters of the algorithm in most cases (*Jones, Mirrazavi & Tamiz, 2002*). Finding the optimal parameter values is a laborious task that requires expertise and knowledge about the algorithm, its parameters, and the problem (*Neumüller et al., 2012*). To determine the appropriate parameter values for the problem, the algorithm must be run many times with different sets of parameters. Tuning problem-specific parameter values is an optimization problem in itself. As the number of parameters increases, the permutations of the parameter sets increase, and parameter optimization of the algorithm becomes more time-consuming. This makes it difficult to use the metaheuristic algorithm when time or other constraints allow only a single run.

The only parameter that needs to be tuned in the FA is the number of neighbours to scan in each step, referred to as $m$ in the Algorithm 1. This parameter has no direct effect on the way solutions are generated or the acceptance process of the generated solutions. Its main effect is on the running time of the algorithm. Increasing the value of this parameter helps to scan the solution space on a global scale by generating more different solutions, but causes the algorithm to finish in a longer time. The fact that the FA has a single basic

parameter makes it easily adaptable to different problems, and parameter optimization processes are quite simple.

## Exploitation and exploration

The main components of any metaheuristic algorithm are exploitation and exploration or intensification and diversification (*Blum & Roli 2003*). The efficient searchability of a metaheuristic algorithm heavily relies on these two components (*Liu et al. 2013*). A widely accepted principle among researchers is that metaheuristic search methods can achieve better performance when an appropriate balance between exploration and exploitation of solutions is achieved (*Xu & Zhang, 2014*). Although several criteria have been proposed in the literature, such as measuring the diversity in the current population, which is recommended for population-based algorithms, there is no definitive way to objectively measure the rate of exploration/exploitation provided in a metaheuristic scheme (*Morales-Castañeda et al., 2020*).

Exploration refers to searching the global solution space by generating different solutions, including solutions that are even worse than the available solutions, while exploitation refers to searching in a local region by exploiting a good solution. Metaheuristic algorithms are expected to use these two mechanisms in a balanced and efficient way to achieve the best results. If there is an imbalance in favor of exploitation, the system may get stuck in the local optimum and not be able to reach the global optimum. On the contrary, if there is an imbalance in favor of exploration, it may turn into a random search and be difficult to converge.

The FA is a trajectory-based algorithm and the balance between exploration and exploitation is exploitation-prone due to the nature of the flood flow it is inspired by. Solutions with better fitness values are always accepted. This can manifest as local (greedy) search steps early in the search process. However, the discovery of better solutions increases the speed value, allowing the search process to continue, and avoiding getting stuck in a local optimum by accepting the bad results that will come in the ongoing process. As mentioned in the "Mathematical model" section the FA has two basic rules for selecting the new solution for the next iteration. While the first rule aims to converge to better solutions by using the information on a good solution in a local area as the exploitation, the second rule aims to avoid getting stuck at local optimums by accepting an even worse solution than the current one as the exploration. These rules provide a balance between exploitation and exploration. The FA balances exploration and exploitation through the speed parameter. The speed parameter of the algorithm, which increases with better solutions, allows the algorithm to accept worse solutions in subsequent iterations and to explore different regions of the solution space. The search mechanism, which is prone to exploitation, can move to other regions by accepting worse solutions after the best solution it has reached locally. The search patterns in the solution space of the benchmark functions of the FA are examined in the "Flood Algorithm Validation" section. It is seen that the FA has a successful search propagation around the local and global minima of the functions.

Selection of the solution to be used for the next iteration is accomplished by comparing the fitness values of the generated solutions to the current solution or to each other. There are no special operators or time-consuming complex position calculations that need to be tailored to the problem. If there is a solution among the neighbours with a better fitness value than the current solution, it is selected; if not, the best solution among the neighbours is selected. This simple structure makes it easy to apply the flooding algorithm to various types of optimization problems, including permutation problems.

## Complexity

This subsection investigates the computational complexity of the proposed algorithm. In order to evaluate the computational complexity of a novel metaheuristic algorithm, the "Big O notation" can be used, which is a mathematical notation that represents the required running time and memory space of an algorithm by considering the growth rate in dealing with different inputs. The main factors that affect the computational complexity of the FA in solving an optimization problem are the number of iterations (*i.e.*, N), the number of neighbour solutions generated in each iteration (*i.e.*, $k_{max}$), and the cost of the problem's objective function (*i.e.*, $O(f_{fitness})$) calculated for each neighbor. In this context, the worst-case computational complexity of the FA is $O(N\ k_{max}\ O(f_{fitness}))$.

The complexity of the fitness function $O(f_{fitness})$ increases depending on the number of decision variables (*i.e.*, D) in the optimization problems. In this context, we can use the number of decision variables as a multiplier instead of the complexity of the fitness function in overall complexity and update the overall complexity as seen in Eq. (9).

$$O(N\ k_{max}\ D). \tag{9}$$

## FLOOD ALGORITHM VALIDATION

To determine the performance of a metaheuristic algorithm, a sufficient number of experiments and case studies should be performed. As part of the verification of the FA, 15 benchmark functions, three engineering design problems and a real-world problem of preparation of an exam seating plan were used. In order to verify the search capability of the FA, GA, SA, and PSO algorithm are executed on the same benchmark function set, engineering design problems and the real-world problem. The results obtained are compared to the FA. These algorithms were chosen because they are the most widely used algorithms in the literature on optimization problems as shown in Fig. 3.

### Benchmark functions

In this subsection, FA is investigated on a set of 15 benchmark functions, eight of which are from the CEC2022 test suite. The range, dimension, type, and formulation of functions are listed in Table 2. The list contains unimodal, multimodal, separable, and non-separable functions with different dimensions. The main goal is to evaluate the performance of the FA by using the difficulty of the different benchmark problems.

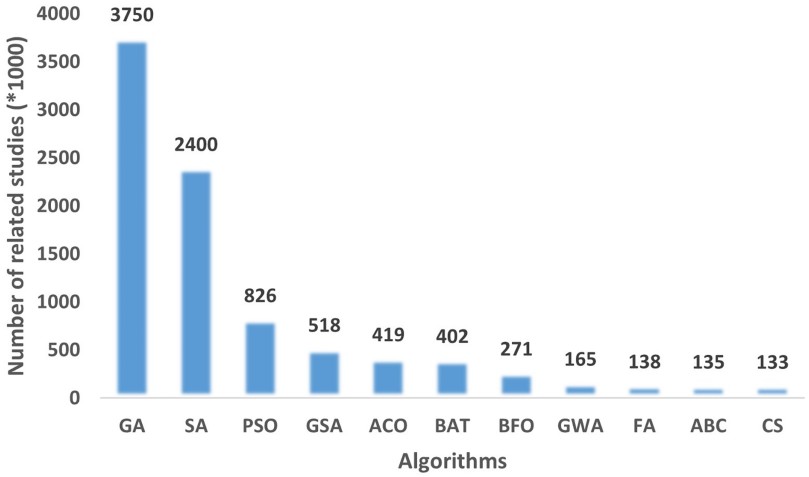

**Figure 3 The number of related articles on Google Scholar for metaheuristic algorithms.**

**Table 2 Benchmark functions used in experiments.**

| Function | Range | D | Type | Formulation |
|---|---|---|---|---|
| Ackley | −32, 32 | 30 | MN | $f(\mathbf{x}) = -20exp(-0.2\sum_{i=1}^{D} x_i^2) - exp(1/d\sum_{i=1}^{D}\cos 2\pi x_i) + exp(1) + 20$ |
| Beale | −4.5, 4.5 | 2 | UN | $f(\mathbf{x}) = (1.5 - x_1 + x_1 x_2)^2 + (2.25 - x_1 + x_1 x_2^2)^2 + (2.625 - x_1 + x_1 x_2^3)^2$ |
| Easom | −100, 100 | 2 | UN | $f(\mathbf{x}) = -\cos x_1\cos x_2 exp(-(x_1 - \pi)^2 - (x_2 - \pi)^2)$ |
| Michalewicz | 0, $\pi$ | 10 | MS | $f(\mathbf{x}) = -\sum_{i=1}^{D}\sin x_i\sin^{20} i x_i^2/\pi$ |
| Quartic | −1.28, 1.28 | 30 | US | $f(\mathbf{x}) = \sum_{i=1}^{D} i x_i^4 + Rand$ |
| Rastrigin | −5.12, 5.12 | 30 | MS | $f(x) = 10D + \sum_{i=1}^{D}[x_i^2 - 10\cos 2\pi x_i]$ |
| Shubert | −10, 10 | 2 | MN | $f(\mathbf{x}) = \sum_{i=1}^{5} i\cos((i+1)x_1 + i)\sum_{i=1}^{5} i\cos((i+1)x_2 + i)$ |
| Six hump camel back | −5, 5 | 2 | MN | $f(\mathbf{x}) = (4 - 2.1x_1^2 + (x_1^4/3))x_1 32 + x_1 x_2 + (-4 + 4x_2^2)x_2^2$ |
| Sphere | −100, 100 | 30 | US | $f(\mathbf{x}) = \sum_{i=1}^{d} x_i^2$ |
| Bent Cigar | −100, 100 | 30 | UN | $f(\mathbf{x}) = x_1^2 + 10^6\sum_{i=2}^{n} x_i^2$ |
| HGBat | −15, 15 | 30 | MN | $f(\mathbf{x}) = \left\|\left(\sum_{i=1}^{D} x_i^2\right)^2 - \left(\sum_{i=1}^{D} x_i\right)^2\right\|^{\frac{1}{2}} + \frac{0.5\sum_{i=1}^{D} x_i^2 + \sum_{i=1}^{D} x_i}{D} + 0.5$ |
| Griewank | −100, 100 | 30 | MN | $f(\mathbf{x}) = \frac{1}{4,000}\sum_{i=1}^{n} x_i^2 - \prod_{i=1}^{n}\cos\left(\frac{x_i}{\sqrt{i}}\right) + 1$ |
| Rosenbrock | −10, 10 | 30 | MN | $f(\mathbf{x}) = \sum_{i=1}^{D-1}\left(100(x_{i+1} - x_i^2)^2 + (x_i - 1)^2\right)$ |
| Levy | −10, 10 | 30 | MN | $f(\mathbf{x}) = \sin^2(\pi x_1) + \sum_{i=1}^{n-1}(x_i - 1)^2[1 + 10\sin^2(\pi x_{i+1})] + (x_D - 1)^2$ |
| Zakharov | −5, 10 | 30 | UN | $f(\mathbf{x}) = \sum_{i=1}^{n} x_i^2 + \left(\frac{1}{2}\sum_{i=1}^{n} i x_i\right)^2 + \left(\frac{1}{2}\sum_{i=1}^{n} i x_i\right)$ |

**Note:**
*D*, dimension; *M*, multimodal; *U*, unimodal; *S*, separable; *N*, non-seperable.

## Experimantal results

For benchmark functions, the FA, GA, SA, and PSO algorithm were run 100 times on a
64-bit computer with 16 GB RAM and Intel Core i5 CPU, with the parameter values given

**Table 3 Parameter values of algorithms.**

| Algorithm | Parameter | Value |
|-----------|-----------|-------|
| FA | Neighbour count | 30 |
| SA | $k_{max}$ | 10,000 |
| PSO | c1 | 2 |
| | c2 | 2 |
| | Inertia weight | 0.8 |
| GA | Crossover rate | 0.8 |
| | Mutation rate | 0.05 |

in Table 3. To ensure the fairness of the experiment, the population-based GA and PSO algorithm were run for 100 generations/cycle on a population of 100 individuals which means 10,000 function evaluations, and the trajectory-based FA and SA were run for 10,000 steps which also means 10,000 function evaluations. The algorithms were compared in terms of fitness value and convergence performance on benchmark functions.

The minimum, maximum, average, and standard deviation of fitness values obtained for each algorithm are given in Table 4. When the results are analyzed, it is seen that the FA produces better results than the other algorithms in 31 out of 60 different comparison areas (especially on multi-dimensional functions), including best, worst, average, and standard deviation for each benchmark function. In addition to these comparisons, the Wilcoxon Ranksum Test (*Wilcoxon, 1992*), a non-parametric statistical test, was used to evaluate the results statistically. The ranks and *p*-values (for statistically significance level at $\alpha = 0.05$) of the results obtained by the FA against the results obtained by the other three algorithms for 15 benchmark functions are presented in Table 5. According to the Wicoxon Ranksum Test results, the FA outperformed the GA in 12 out of 15 functions, the PSO in 11 out of 15 functions, and the SA in nine out of 15 functions.

The convergence comparison graphs of the algorithms are given in Fig. 4. These graphs clearly show the better final result as well as the better convergence trend of the proposed FA compared to GA, SA, and PSO.

Another investigation on the benchmark functions was the search patterns of the flood algorithm. The search patterns of the flood algorithm on the solution space of the benchmark functions are given in Figs. 5 and 6. In the graphs in Figs. 5 and 6, the number of neighbours, which is the only parameter of the FA that needs to be determined in advance, is considered as 30 and the benchmark functions are considered as two-dimensional. It can be seen that the flood algorithm has a successful search propagation around the local and global minimum points of the functions. When the effect of the number of neighbours on the search propagation is examined, it is seen that as the number of neighbours increases, as expected the search propagation becomes sharper towards the regions with minimum points, as seen in Fig. 7. Increasing the number of neighbours leads to better results, but increases the execution time of the algorithm at the same rate. The
**Table 4 Comparative results of FA with GA, PSO, and SA (bolded numbers represent the best values).**

| Function | | FA | GA | PSO | SA |
|---|---|---|---|---|---|
| Ackley | Best | 0.630264721 | **0.030177472** | 6.037359303 | 0.615644664 |
| | Worst | **1.081308917** | 9.852174721 | 15.00098476 | 1.890181228 |
| | Mean | **0.840619763** | 4.827023494 | 9.887241081 | 1.21125692 |
| | StdDev | **0.102621848** | 2.458913203 | 1.485300744 | 0.284202246 |
| Beale | Best | 8.1218E−09 | 1.50587E−05 | **5.32812E−15** | 2.57522E−06 |
| | Worst | 0.763954554 | **0.017210748** | 0.735040539 | 9.542461754 |
| | Mean | 0.243985461 | **0.004337761** | 0.007350405 | 0.646968955 |
| | StdDev | 0.357440055 | **0.003958047** | 0.073504054 | 1.522015766 |
| Easom | Best | −0.999984745 | −0.999733691 | **−1** | −0.999989072 |
| | Worst | −8.0573E−05 | −6.48809E−05 | **−0.999999997** | −5.71945E−09 |
| | Mean | −0.559012012 | −0.486018129 | **−1** | −0.895178564 |
| | StdDev | 0.497941702 | 0.336835486 | **4.53634E−10** | 0.295969316 |
| Michalewicz | Best | −9.655901223 | −5.42823829 | −9.035137456 | **−9.657388029** |
| | Worst | **−9.628734725** | −3.154691032 | −5.024521985 | −9.590041807 |
| | Mean | **−9.644992901** | −4.157626983 | −6.958815078 | −9.641043957 |
| | StdDev | **0.004907447** | 0.387775307 | 0.805336095 | 0.013817348 |
| Quatric | Best | **7.11854E−07** | 1.04821E−06 | 0.033055868 | 0.000210782 |
| | Worst | **1.43489E−05** | 3.265309166 | 1.620159254 | 0.001263889 |
| | Mean | **6.29445E−06** | 0.41820625 | 0.36219887 | 0.000661741 |
| | StdDev | **2.80352E-06** | 0.719218 | 0.287628592 | 0.000220706 |
| Rastrigin | Best | 0.924309115 | **0.044107217** | 70.95627643 | 1.041741143 |
| | Worst | **2.358124187** | 38.83575726 | 197.9559435 | 5.304766346 |
| | Mean | **1.526722464** | 13.23619498 | 116.5297821 | 2.621956417 |
| | StdDev | **0.317304598** | 8.777635217 | 24.5573117 | 0.834267081 |
| Shubert | Best | −186.7308897 | −186.7248143 | **−186.7309088** | −186.7309027 |
| | Worst | **−186.7163675** | −177.2558732 | −184.7264808 | −186.6764707 |
| | Mean | **−186.7289436** | −185.1291998 | −186.7108645 | −186.7263416 |
| | StdDev | **0.002344217** | 1.682359271 | 0.200442803 | 0.007494791 |
| Sixhump camelback | Best | −1.031628452 | −1.031470545 | **−1.031628453** | −1.031628351 |
| | Worst | −1.031566413 | −0.980048867 | **−1.031628452** | −1.031261734 |
| | Mean | −1.031617878 | −1.022595193 | **−1.031628452** | −1.031582382 |
| | StdDev | 1.18116E−05 | 0.009334517 | **4.09659E−12** | 6.57447E−05 |
| Sphere | Best | **1.88159E−05** | 0.005840427 | 622.3560198 | 1.803937595 |
| | Worst | **0.007377774** | 4,378.884757 | 8,637.558451 | 5.282826775 |
| | Mean | **0.001233648** | 798.8495789 | 2,180.258882 | 3.533097216 |
| | StdDev | **0.00141772** | 901.084707 | 1142.749793 | 0.734131596 |
| Bent Cigar | Best | 1,662,639.001 | **52,208.38673** | 306,655,833.9 | 1,852,577.25 |
| | Worst | **11,315,660.86** | 3,904,504,464 | 3,968,514,125 | 12,610,546.22 |
| | Mean | **5,132,787.392** | 718,352,211.1 | 1,836,472,254 | 5,456,679.957 |
| | StdDev | **1,991,141.08** | 705,159,229.7 | 790,638,188.1 | 2,221,453.951 |
| HGBat | Best | **0.273985563** | 0.580532757 | 11.37543708 | 0.309354292 |

| Function | | FA | GA | PSO | SA |
|---|---|---|---|---|---|
| | Worst | 1.490868581 | 121.6640925 | 115.6925306 | **1.248709457** |
| | Mean | 0.680847783 | 24.20371437 | 45.2520308 | **0.622166672** |
| | StdDev | 0.355691216 | 25.01670369 | 20.00026061 | **0.292947893** |
| Griewank | Best | 0.070946968 | **0.010918879** | 1.181959635 | 0.171126239 |
| | Worst | **0.444453243** | 2.303547801 | 2.288771959 | 0.736621856 |
| | Mean | **0.234341341** | 0.717726596 | 1.511488535 | 0.387217392 |
| | StdDev | **0.069850661** | 0.415042013 | 0.228033621 | 0.092241659 |
| Rosenbrock | Best | **4.984522855** | 29.01762542 | 75.22963067 | 7.060738999 |
| | Worst | **141.1286931** | 373.1991982 | 236.5369623 | 141.876423 |
| | Mean | **57.07679761** | 69.06191623 | 133.5588939 | 62.7984998 |
| | StdDev | **31.50056937** | 50.00424247 | 35.83975214 | 37.44949298 |
| Levy | Best | 0.030788386 | 2.710117694 | 2.552523385 | **0.008846786** |
| | Worst | 0.141272967 | 9.097557631 | 29.71448051 | **0.057414966** |
| | Mean | 0.069551636 | 4.171587424 | 10.68035128 | **0.026914693** |
| | StdDev | 0.022433369 | 1.281362725 | 5.429156246 | **0.010534223** |
| Zakharov | Best | 155.825532 | **0.046912753** | 236.0268778 | 166.4557122 |
| | Worst | 538.0403138 | **81.9534772** | 945.6585919 | 565.4200336 |
| | Mean | 332.1240272 | **26.33974406** | 494.4301908 | 339.3193892 |
| | StdDev | 75.9756031 | **17.76458461** | 147.9450299 | 76.08398062 |

**Table 5 Wilcoxon rank-sum test results.**

| Function | FA *vs.* GA | | | FA *vs.* PSO | | | FA *vs.* SA | | |
|---|---|---|---|---|---|---|---|---|---|
| | $R^+$ | $R^-$ | *p*-value | $R^+$ | $R^-$ | *p*-value | $R^+$ | $R^-$ | *p*-value |
| Ackley | 14,370 | 5,730 | **4.86E−26** | 15,050 | 5,050 | **2.56E−34** | 13,972 | 6,128 | **9.54E−22** |
| Beale | 11,832 | 8,268 | **1.34E−05** | 5,118 | 14,982 | 1.95E−33 | 12,693 | 7,407 | **1.07E−10** |
| Easom | 10,705 | 9,395 | 1.10E−01 | 5,050 | 15,050 | 2.56E−34 | 9,571 | 10,529 | 2.42E−01 |
| Michalewicz | 15,050 | 5,050 | **2.56E−34** | 15,050 | 5,050 | **2.56E−34** | 10,259 | 9,841 | 6.10E−01 |
| Quatric | 14,771 | 5,329 | **8.89E−31** | 15,050 | 5,050 | **2.56E−34** | 15,050 | 5,050 | **2.56E−34** |
| Rastrigin | 14,621 | 5,479 | **5.88E−29** | 15,050 | 5,050 | **2.56E−34** | 14,095 | 6,005 | **4.97E−23** |
| Shubert | 15,045 | 5,055 | **2.98E−34** | 5,150 | 14,950 | 5.02E−33 | 11,162 | 8,938 | **6.60E−03** |
| Six hump camel back | 15,050 | 5,050 | **2.56E−34** | 5,050 | 15,050 | 2.56E−34 | 12,607 | 7,493 | **4.20E−10** |
| Sphere | 15,048 | 5,052 | 2.72E−34 | 15,050 | 5,050 | **2.56E−34** | 15,050 | 5,050 | **2.56E−34** |
| Bent Cigar | 14,758 | 5,342 | **1.29E−30** | 15,050 | 5,050 | **2.56E−34** | 10,475 | 9,625 | 3.00E−01 |
| HGBat | 14,609 | 5,491 | **8.18E−29** | 15,050 | 5,050 | **2.56E−34** | 10,056 | 10,044 | 9.89E−01 |
| Griewank | 13,985 | 6,115 | **7.01E−22** | 15,050 | 5,050 | **2.56E−34** | 14,241 | 5,859 | **1.33E−24** |
| Rosenbrock | 10,656 | 9,444 | 1.39E−01 | 14,743 | 5,357 | **1.96E−30** | 10,338 | 9,762 | 4.82E−01 |
| Levy | 15,050 | 5,050 | **2.56E−34** | 15,050 | 5,050 | **2.56E−34** | 5,250 | 14,850 | **9.26E−32** |
| Zakharov | 5,050 | 15,050 | **2.56E−34** | 13,394 | 6,706 | **3.10E−16** | 10,298 | 9,802 | 5.45E−01 |

**Note:**
Bold values represent the values better than the level of significance $\alpha = 0.05$.

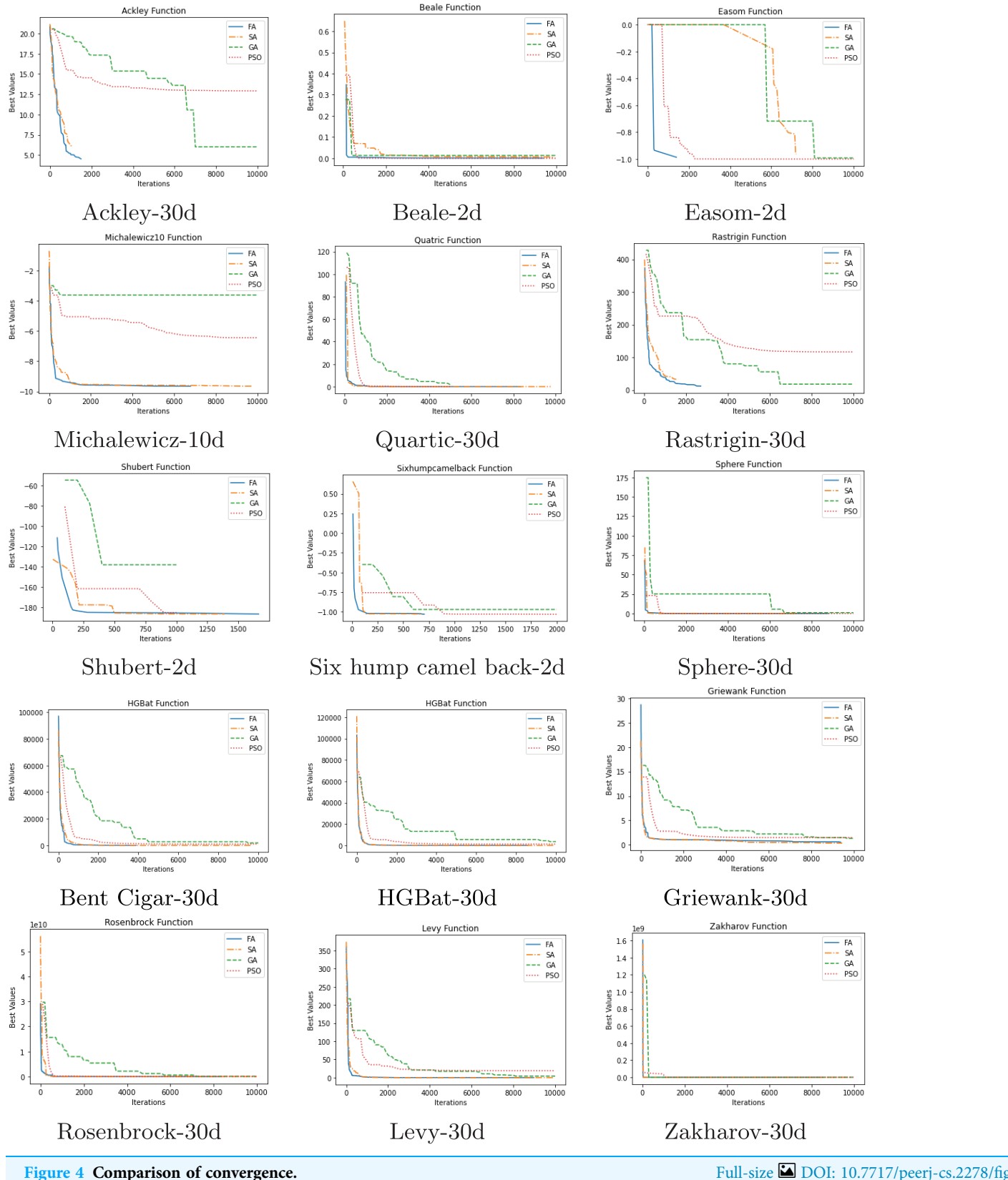

**Figure 4  Comparison of convergence.**

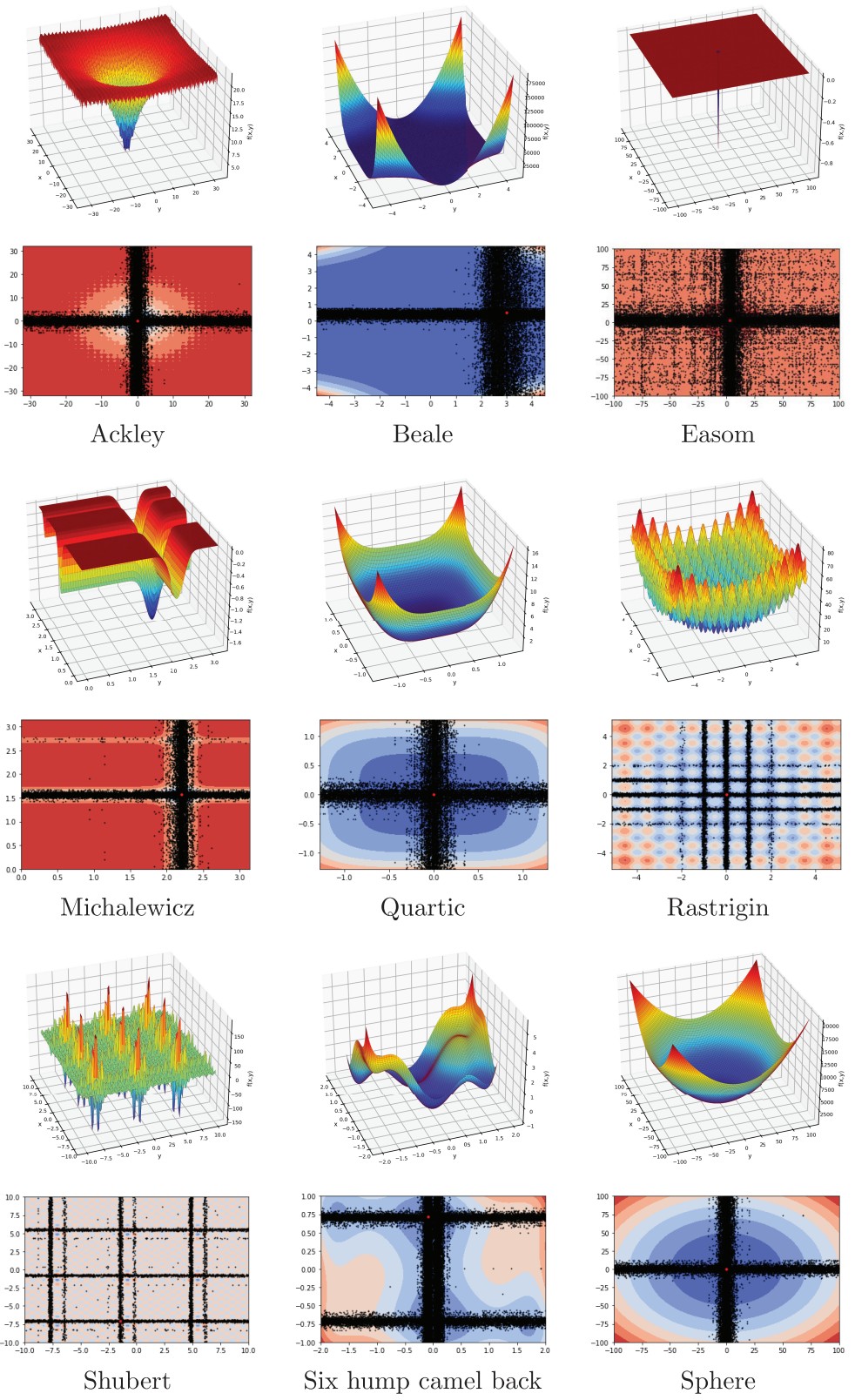

**Figure 5 Flood algorithm search history with 30 neighbours-1.**

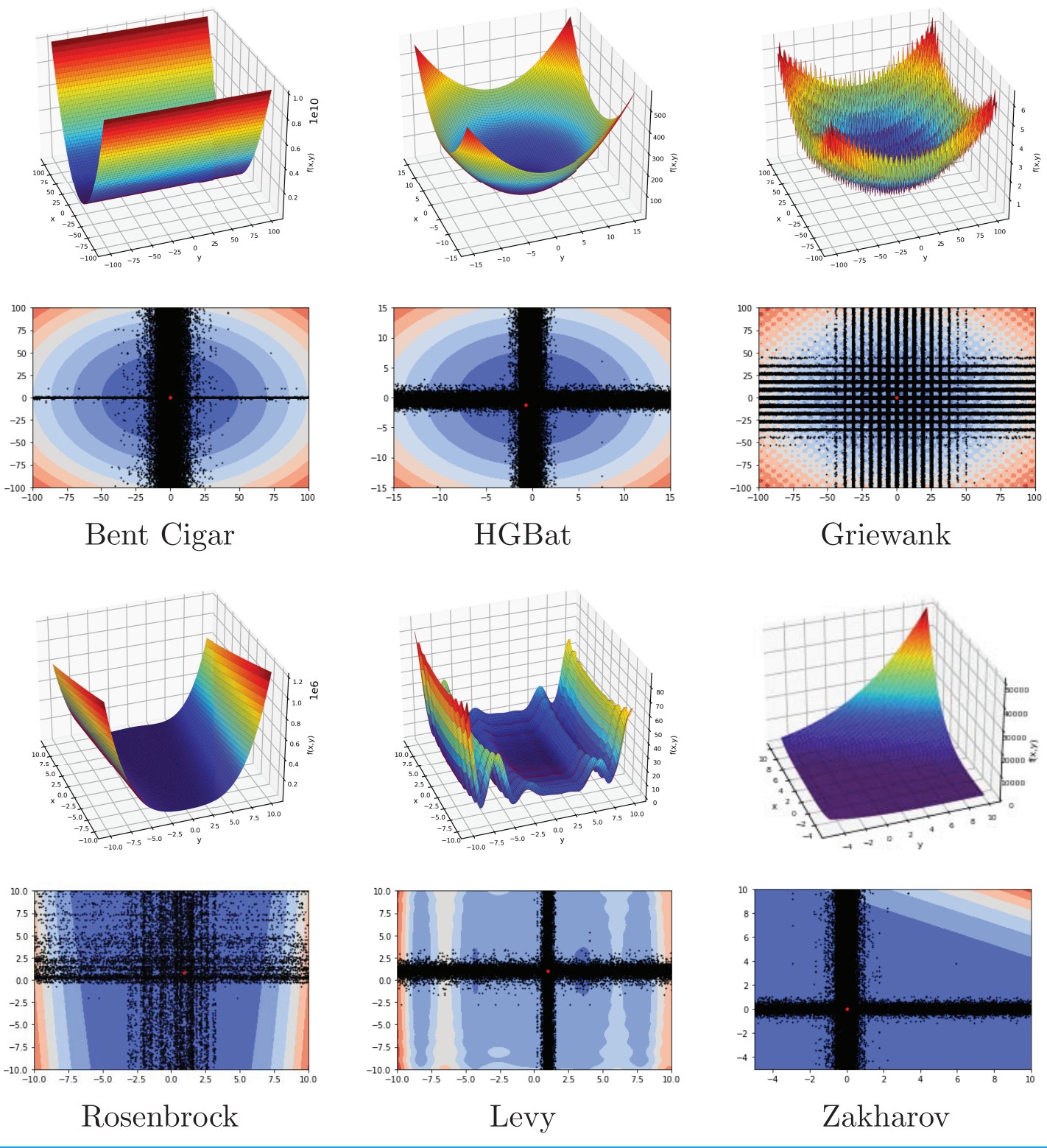

**Figure 6 Flood algorithm search history with 30 neighbours-2.**

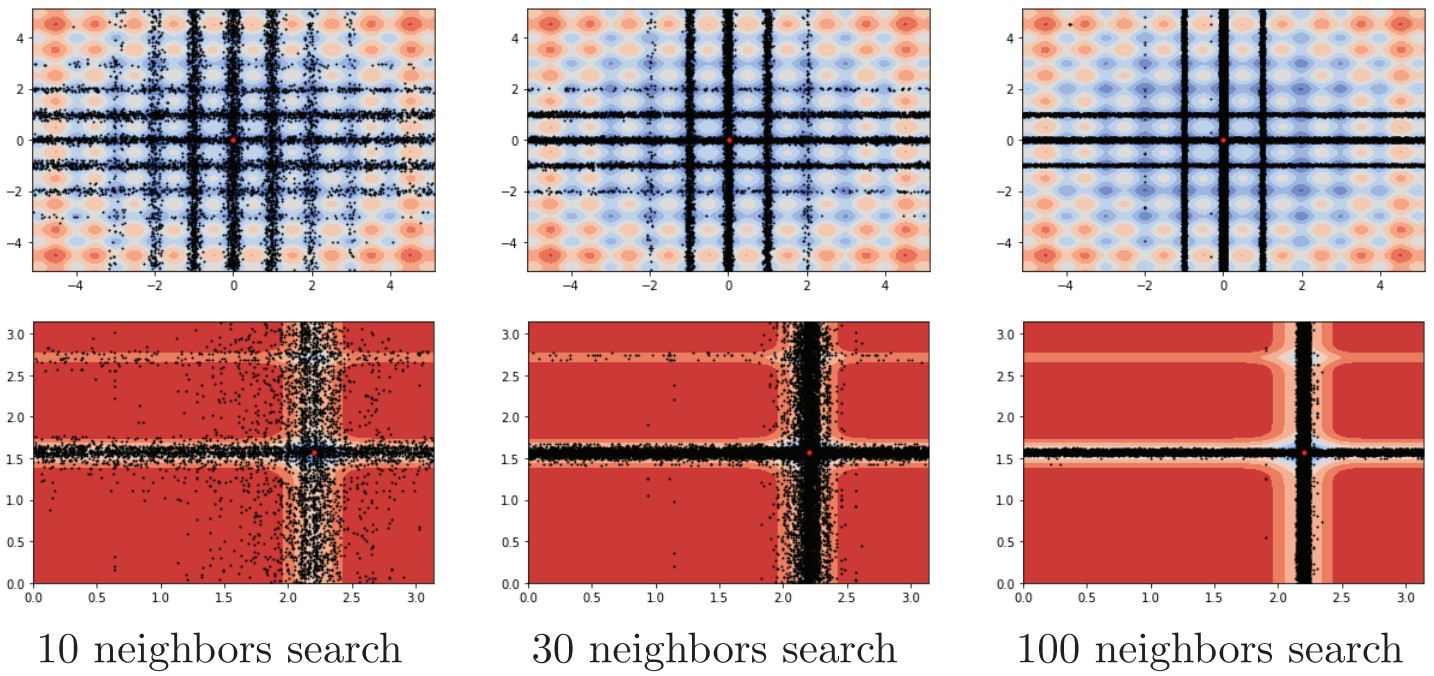

10 neighbors search    30 neighbors search    100 neighbors search

**Figure 7 Search history of different neighbour rates.**

graphs in Fig. 7 show the results of the analysis on two benchmark functions (Rastrigin and Michalewicz) for three different numbers of neighbours (10, 30 and 100).

### Engineering design problems

The FA has been applied to three of the classic engineering design problems that have been widely studied in the literature. These three engineering problems are the three-bar truss design problem, the pressure vessel design problem and the tension/compression spring design problem. The objective function and constraints of the three-bar truss design problem are described by the Formula (10).

$$f(\mathbf{x}) = (2\sqrt{2}x_1 + x_2) \ \text{X} \ l$$

with restrictions :

$$g_1(\mathbf{x}) = \frac{2\sqrt{2}x_1 + x_2}{2\sqrt{2}x_1^2 + 2x_1x_2}p - \sigma \leq 0$$

$$g_2(\mathbf{x}) = \frac{x_2}{2\sqrt{2}x_1^2 + 2x_1x_2}p - \sigma \leq 0 \tag{10}$$

$$g_3(\mathbf{x}) = \frac{1}{2\sqrt{2}x_2 + x_1}p - \sigma \leq 0$$

$$0 \leq x_1, x_2 \leq 1; \quad l = 100; \quad p = 2; \quad \sigma = 2.$$

The objective function and constraints of the pressure vessel design problem are described by the Formula (11).

$$f(\mathbf{x}) = 0.6224\ x_1\ x_3\ x_4\ +\ 1.7781\ x_2\ x_3^2\ +\ 3.1661\ x_1^2\ x_4\ +\ 19.84\ x_1^2\ x_3$$

with restrictions :

$$g_1(\mathbf{x}) = -x_1\ +\ 0.0193\ x_3\ \leq\ 0$$

$$g_2(\mathbf{x}) = -x_2\ +\ 0.00954\ x_3\ \leq\ 0 \tag{11}$$

$$g_3(\mathbf{x}) = -\pi\ x_3^2\ x_4\ -\ \frac{4}{3}\ \pi\ x_3^3\ +\ 1296000\ \leq\ 0$$

$$g_4(\mathbf{x}) = x_4\ -\ 240\ \leq\ 0$$

$$0\ \leq\ x_1, x_2\ \leq\ 99;\quad 10\ \leq\ x_3, x_4\ \leq\ 200.$$

The objective function and constraints of the tension/compression spring design problem are described by the Formula (12).

$$f(\mathbf{x}) = (x_3\ +\ 2)\ x_2\ x_1^2$$

with restrictions :

$$g_1(\mathbf{x}) = 1\ \frac{x_2^3\ x_3}{71{,}785\ x_1^4} \leq\ 0$$

$$g_2(\mathbf{x}) = \frac{4\ x_2^2\ -\ x_1\ x_2}{12{,}566\ (x_2\ x_1^3\ -\ x_1^4)}\ +\ \frac{1}{5{,}108\ x_1^2}\ -\ 1\ \leq\ 0 \tag{12}$$

$$g_3(\mathbf{x}) = 1\ -\ \frac{140.45\ x_1}{x_2^2\ x_3}\ \leq\ 0$$

$$g_4(\mathbf{x}) = \frac{x_1\ +\ x_2}{1.5}\ -\ 1\ \leq\ 0$$

$$0.05\ \leq\ x1\ \leq\ 2;\quad 0.25\ \leq\ x1\ \leq\ 1.3;\quad 2\ \leq\ x1\ \leq\ 15;$$

### Experimantal results

To solve these problems, all of the compared algorithms were run 100 times. The minimum, maximum, average, and standard deviation of fitness values obtained for each algorithm are given in Table 6. As seen in Table 6, the FA produces better results than the other algorithms in nine out of 12 comparison areas for three design problems.

## Exam seating problem

In addition to the benchmark functions and engineering design problems, we tested the FA on the exam seating problem of a university as a real-world problem with the other three metaheuristics. In an examination session, one of the main concerns is the successful distribution of seats in examination halls (*Chaki & Anirban, 2016*). In this real-world problem, the goal is to minimize the number of students taking the same exam and sitting side by side or back to back in an exam session by using the minimum number of classrooms (without leaving any empty seats). The solutions must include all students taking an exam in the session, and each student must be included in the solutions only once. Under these conditions, the exam seating problem becomes an NP-complete permutation problem similar to the Travelling Salesman Problem (TSP), where the student

**Table 6 Comparative results of FA with GA, PSO, and SA (bolded numbers represent the best values).**

| Problems | | FA | GA | PSO | SA |
|---|---|---|---|---|---|
| Three-bar truss | Min | 159.1 | 159.1336 | **159.099** | 159.1129 |
| | Max | **159.181** | 165.5675 | 160.329 | 279.4735 |
| | Average | 159.124 | 161.2898 | **159.1129** | 184.5323 |
| | StdDev | **0.014192** | 1.258395 | 0.123407 | 31.61629 |
| Pressure vessel | Min | 1,253.023 | 3,562.545 | **1,089.464** | 1,857.01 |
| | Max | **1,755.822** | 63,852.54 | 541,954.3 | 5,883.714 |
| | Average | **1,471.384** | 24,629.07 | 33,416.63 | 4,525.2 |
| | StdDev | **95.21283** | 14,010.81 | 91,308.28 | 782.6464 |
| Tension spring | Min | **0.012689** | 0.013098 | 0.012667 | 0.012706 |
| | Max | **0.014842** | 0.044247 | 0.042496 | 0.0222 |
| | Average | **0.012999** | 0.018458 | 0.017713 | 0.016947 |
| | StdDev | **0.000429** | 0.004857 | 0.005221 | 0.002598 |

**Table 7 Structure of solution vector.**

**Seat indexes**

| | 1 | 2 | 3 | 4 | 5 | 6 | 7 | 8 | 9 | 10 | |
|---|---|---|---|---|---|---|---|---|---|---|---|
| Solution -> Student numbers | $S_2$ | $S_8$ | $S_6$ | $S_{10}$ | $S_4$ | $S_7$ | $S_1$ | $S_5$ | $S_3$ | $S_9$ | ... |

seating order corresponds to the order of cities to visit. In the modeling phase of the problem, the structure of the possible solutions is considered as a vector as shown in Table 7, where the indices represent the seats and the values in the indices represent the students. A traceability matrix, seen in Table 8, was created representing all possible classrooms and seat locations, including side-by-side and back-to-back seat location neighbours, allowing access to seat location neighbours through index information in the solution vector. Eleven exam session data from one exam period of the university were used as the data set. For each exam session, a student-course matrix containing the student numbers and the course information was created and access to the course information was provided through the student number.

### Experimantal results

For the exam seating problem, the FA, SA, GA, and PSO algorithm were run 50 times for each exam session with the parameter values given in Table 3. The number of courses and number of students in the exam sessions are given in Table 9.

The minimum, maximum, average, and standard deviation of fitness values obtained for each algorithm are given in Table 10. A seen in Table 10, the FA produces better results than the other algorithms in 42 out of 44 comparison areas for 11 exam sessions. In

**Table 8 Location traceability matrix view.**

| # Of seat | Classroom | Name | Row | Column | Side | Back |
|---|---|---|---|---|---|---|
| 1 | 1 | D-301 | 1 | 1 | 2 | 6 |
| 2 | 1 | D-301 | 1 | 2 | −1 | 7 |
| 3 | 1 | D-301 | 1 | 3 | −1 | 8 |
| 4 | 1 | D-301 | 1 | 4 | 5 | 9 |
| 5 | 1 | D-301 | 1 | 5 | −1 | 10 |
| 6 | 1 | D-301 | 2 | 1 | 7 | 11 |

Note:
A total of −1 in *side* or *back* columns means that there is no adjacent seat location in that direction.

**Table 9 Course and student numbers of exam sessions.**

**Exam sessions**

| | 1 | 2 | 3 | 4 | 5 | 6 | 7 | 8 | 9 | 10 | 11 |
|---|---|---|---|---|---|---|---|---|---|---|---|
| # Of courses | 12 | 8 | 9 | 9 | 8 | 11 | 8 | 4 | 4 | 7 | 13 |
| # Of students | 827 | 830 | 821 | 869 | 775 | 584 | 712 | 434 | 643 | 833 | 879 |

**Table 10 Comparative results of FA with GA, PSO, and SA (bolded numbers represent the best values).**

| Sessions | | FA | GA | PSO | SA |
|---|---|---|---|---|---|
| 1 | Min | **1** | 11 | 48 | **1** |
| | Max | **6** | 27 | 87 | 10 |
| | Average | **3.42** | 19.34 | 67.22 | 5.34 |
| | StdDev | **1.485851642** | 4.697416 | 11.57635 | 2.076201398 |
| 2 | Min | **0** | 16 | 53 | 1 |
| | Max | **9** | 38 | 92 | 11 |
| | Average | **3.14** | 28.12 | 74.18 | 5.72 |
| | StdDev | **2.060414066** | 6.610567 | 12.12804 | 2.138471703 |
| 3 | Min | **1** | 17 | 46 | 2 |
| | Max | **7** | 39 | 84 | 11 |
| | Average | **3.48** | 27.3 | 67.22 | 6.92 |
| | StdDev | **1.606618962** | 6.609363 | 11.3034 | 2.257323682 |
| 4 | Min | **1** | 25 | 55 | 2 |
| | Max | **7** | 47 | 94 | 12 |
| | Average | **3.98** | 36.4 | 76.76 | 6.64 |
| | StdDev | **1.634949915** | 7.645193 | 11.51496 | 2.310225716 |
| 5 | Min | **4** | 11 | 64 | 6 |
| | Max | **17** | 36 | 102 | 21 |
| | Average | **9.88** | 24.9 | 84.6 | 13.24 |
| | StdDev | **2.760065069** | 7.707007 | 9.822922 | 2.924631496 |

| Table 10 (continued) | | | | | |
|---|---|---|---|---|---|
| **Sessions** | | **FA** | **GA** | **PSO** | **SA** |
| 6 | Min | **0** | 3 | 46 | **0** |
| | Max | **2** | 17 | 82 | **2** |
| | Average | **0.12** | 10.26 | 62.78 | 0.32 |
| | StdDev | **0.385449645** | 4.503106 | 11.3969 | 0.512695956 |
| 7 | Min | **0** | 36 | 63 | 4 |
| | Max | **11** | 72 | 101 | 15 |
| | Average | **6.02** | 53.4 | 83.36 | 8.82 |
| | StdDev | 2.453568829 | 10.66752 | 10.54197 | **2.404842054** |
| 8 | Min | **39** | 54 | 87 | 41 |
| | Max | **60** | 109 | 148 | **60** |
| | Average | 50.3 | 83.26 | 113.06 | **50.2** |
| | StdDev | **4.315373562** | 17.44778 | 21.67643317 | 4.412412918 |
| 9 | Min | **51** | 79 | 84 | 54 |
| | Max | 77 | 138 | 160 | 82 |
| | Average | **60.98** | 122.38 | 120.3 | 68.78 |
| | StdDev | **6.267799145** | 18.00441 | 22.10817 | 6.42472901 |
| 10 | Min | **1** | 27 | 60 | 3 |
| | Max | **13** | 63 | 99 | 18 |
| | Average | **7.08** | 46.26 | 79.04 | 10.08 |
| | StdDev | **2.671390374** | 11.43894 | 12.4249 | 3.439506372 |
| 11 | Min | **0** | 6 | 69 | 1 |
| | Max | **5** | 19 | 106 | 9 |
| | Average | **2.06** | 12.94 | 87.34 | 3.92 |
| | StdDev | **1.300078491** | 3.935241 | 11.93795 | 2.17443401 |

addition to these comparisons, the Wilcoxon rank-sum test was used to evaluate the results statistically. The ranks and $p$-values (for statistically significance level at $\alpha = 0.05$) of the results obtained by the FA against the results obtained by the other three algorithms for exam seating problem are presented in Table 11. According to the Wilcoxon rank-sum test results, the FA outperformed the GA and PSO in 11 out of 11 exam sessions and the SA in 10 out of 11 exam sessions.

In summary, the experimental results clearly show that FA significantly outperforms the other three algorithms on benchmark functions, engineering design problems, and the problem of preparing an exam seating plan. It produces very successful results, especially when the solutions are multidimensional (with a large number of variables). The success in mean, standard deviation and, Wilcoxon ranks-sum test values shows the stability and robustness of FA. The appearance of convergence and search patterns show that FA can successfully scan the solution space and converge to the global optimum.

**Table 11 Wilcoxon rank-sum test results.**

| | FA *vs.* GA | | | FA *vs.* PSO | | | FA *vs.* SA | | |
|---|---|---|---|---|---|---|---|---|---|
| Session | R$^+$ | R$^-$ | *p*-value | R$^+$ | R$^-$ | *p*-value | R$^+$ | R$^-$ | *p*-value |
| 1 | 3,775 | 1,275 | 5.82E−18 | 3,775 | 1,275 | 5.93E−18 | 3,171.5 | 1,878.5 | 6.29E−06 |
| 2 | 3,775 | 1,275 | 6.10E−18 | 3,775 | 1,275 | 6.14E−18 | 3,309.5 | 1,740.5 | 4.83E−08 |
| 3 | 3,775 | 1,275 | 5.68E−18 | 3,775 | 1,275 | 5.77E−18 | 3,481.0 | 1,569.0 | 3.27E−11 |
| 4 | 3,775 | 1,275 | 5.90E−18 | 3,775 | 1,275 | 5.96E−18 | 3,312.5 | 1,737.5 | 4.29E−08 |
| 5 | 3,700 | 1,350 | 4.94E−16 | 3,775 | 1,275 | 6.34E−18 | 3,276.0 | 1,774.0 | 1.96E−07 |
| 6 | 3,775 | 1,275 | 1.57E−19 | 3,775 | 1,275 | 1.62E−19 | 2,770.0 | 2,280.0 | 1.52E−02 |
| 7 | 3,775 | 1,275 | 6.26E−18 | 3,775 | 1,275 | 6.27E−18 | 3,265.0 | 1,785.0 | 2.86E−07 |
| 8 | 3,746 | 1,304 | 3.75E−17 | 3,775 | 1,275 | 6.64E−18 | 2,480.0 | 2,570.0 | 7.58E−01 |
| 9 | 3,775 | 1,275 | 6.83E−18 | 3,775 | 1,275 | 6.84E−18 | 3,301.5 | 1,748.5 | 8.43E−08 |
| 10 | 3,775 | 1,275 | 6.52E−18 | 3,775 | 1,275 | 6.53E−18 | 3,147.5 | 1,902.5 | 1.65E−05 |
| 11 | 3,775 | 1,275 | 4.95E−18 | 3,775 | 1,275 | 5.14E−18 | 3,170.0 | 1,880.0 | 6.22E−06 |

**Note:**
Bold values represent the values better than the level of significance $\alpha = 0.05$.

# CONCLUSION

Optimization problems are problems that we encounter in many different fields such as engineering, finance, and health. It is often not feasible to produce solutions to these problems in polynomial time using deterministic algorithms, due to reasons such as the size of the solution set increasing exponentially according to the number of variables in the objective function and/or the solution space not being convex. In recent decades, many studies have been carried out proposing metaheuristic algorithms that produce intuition-based solutions to solve these optimization problems within an acceptable time frame. The works are largely inspired by natural phenomena. The applications of these highly flexible algorithms in the modeling of complex optimization problems in the engineering world are increasing. In this article, a new meta-heuristic optimization algorithm, called the Flood Algorithm, is proposed. FA is a trajectory-based algorithm inspired by the movement of flood waters on the ground. By scanning the neighbouring points of a randomly determined starting point, when it finds a neighbour with a better fitness value than itself, it passes through that point (modeling the flow of water towards lower points), and when a predetermined number of neighbours are scanned and a neighbour with a better fitness value cannot be found, it passes between neighbours. It is designed to scan the solution space over the point with the best fitness value (modeling the passage of water overflowing from the weakest point of the obstacles it encounters). Experimental results show that the proposed algorithm has stronger search and convergence ability than GA, SA and PSO in most of the comparisons. Currently, the authors are working to compare FA with a larger number of metaheuristic algorithms on a larger pool of problems. The authors look forward to seeing applications of FA on different real-world problems in the future.

### Funding

The authors received no funding for this work.

### Competing Interests

The authors declare that they have no competing interests.

### Author Contributions

- Ramazan Ozkan conceived and designed the experiments, performed the experiments, analyzed the data, performed the computation work, prepared figures and/or tables, authored or reviewed drafts of the article, and approved the final draft.
- Ruya Samli conceived and designed the experiments, analyzed the data, authored or reviewed drafts of the article, and approved the final draft.

### Data Availability

Source code files including "bf" in their names are the source codes of algorithms for benchmark functions. Source code files including "esp" in their names are the source codes of algorithms for exam seating problem. Adjacency and students excel files are the dataset for exam seating problem. Results files are the raw results of algorithms for benchmark functions and exam seating problem. The source code and data is available in the Supplemental Files.

### Supplemental Information

Supplemental information for this article can be found online at http://dx.doi.org/10.7717/peerj-cs.2278#supplemental-information.

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
