# Peer review of "Flood algorithm: a novel metaheuristic algorithm for optimization problems"

_PeerJ Computer Science, doi:10.7717/peerj-cs.2278_

## Round 0.1 · original submission · Major Revisions

· Academic Editor

Major Revisions

Dear authors,

Thank you for your submission. Your article has not been recommended for publication in its current form. However, we do encourage you to address the concerns and criticisms of the reviewers and resubmit your article once you have updated it accordingly.

Best wishes,

**Language Note:** The review process has identified that the English language must be improved. PeerJ can provide language editing services - please contact us at [email protected] for pricing (be sure to provide your manuscript number and title). Alternatively, you should make your own arrangements to improve the language quality and provide details in your response letter. – PeerJ Staff

Reviewer 1 ·

Basic reporting

1. The paper's review of existing research is outdated and needs to be updated with more recent studies.
2. It would be helpful to include mathematical explanations and equations to make the methodology clearer.
3. The flood algorithm concept needs better explanation, possibly with pictures and step-by-step instructions.
4. Consider adding a section that explains the theory in simpler terms for readers who may not be experts in the field. This will make the paper more accessible to a wider audience.

Experimental design

1. The identified problems were characterized by single objectives and relatively straightforward solutions, which may not fully capture the complexity of real-world scenarios.
2. It is essential for the problems addressed to exhibit greater rigor and complexity to adequately assess the performance and robustness of the algorithm under consideration.
3. Experimental designs should aim for multidisciplinary approaches, incorporating diverse perspectives and methodologies to provide a more comprehensive understanding of the algorithm's capabilities and limitations.

Validity of the findings

1. The paper lacks statistical tests to validate the results obtained.
2. Complexity sensitivity is not discussed in detail in the analysis.

Additional comments

The overall idea is commendable, but it should be updated with more recent papers to enhance relevance and accuracy. Additionally, improvements in the writing style could further enhance the paper's quality.

Cite this review as

Reviewer 2 ·

Basic reporting

In this study, the authors developed a new metaheuristic optimization algorithm inspired by the flow of water on Earth. Completing the deficiencies I see in this study is extremely important for the study to reach sufficient maturity.
1-In the article, the authors mentioned the genetic algorithm, particle swarm optimization algorithm, and heat treatment algorithm and stated that the algorithm that gives the best approximate solution may change in every real-world problem. Then, metaheuristic optimization algorithms need to be categorized according to the natural phenomenon they are inspired by. Like light-based, sports-based, math-based. It is necessary to conduct a literature review and add relevant studies.
2-The contributions of FA, a swarm-based method, to the literature should be given in the introduction section.
3-Figure 3 should be removed.

Experimental design

4-Performing The Wilcoxon Rank-Sum test will help the article mature.
5-At least 5 or 6 of the CEC 2022 benchmark functions should be selected and applied to the FA algorithm under equal conditions.

Validity of the findings

6-FA's equations for convergence to global solutions and avoidance of local solutions do not exist. In addition, how to create candidate solutions that are generated randomly and spread over the search space is not given in the equations. Please give the equations of the formulas used for all steps in FA.

Cite this review as

---

## Round 0.2 · Major Revisions

· Academic Editor

Major Revisions

Dear authors,

Thank you for submitting the revised article. Feedback from the reviewers is now available. Although one reviewer accepts the article, it is not recommended that your article be published in its current format by the other reviewer. We strongly recommend that you address the issues raised by this reviewer and resubmit your revised paper after making the necessary changes and additions.

Best wishes,

Reviewer 1 ·

Basic reporting

The introduction of the Flood Algorithm (FA) adds a new perspective to the field of metaheuristic algorithms, drawing inspiration from natural flood processes. This novel approach is commendable and offers potential advantages in solving optimization problems. However, there are several areas where the study could be improved to enhance its validity and reliability.

Expand Experimental Testing:
The current experimental testing, while valuable, is limited. To fully establish the efficacy of the FA, it should be tested on a broader range of benchmark problems, including those from recent literature. This would provide a more comprehensive evaluation of its performance across different problem types and complexities.

Clear Explanation of Exploitation and Exploration:
The paper should include a more detailed explanation of how the FA balances exploitation and exploration. These are critical components of any metaheuristic algorithm, and a clear description of how FA handles these aspects will help in understanding its working mechanism and effectiveness.

Statistical Validation:
It is crucial to perform statistical tests to validate the results. Tests such as ANOVA (Analysis of Variance) and the Wilcoxon rank-sum test should be conducted to determine the statistical significance of the results. These tests will help in establishing whether the observed differences in performance between FA and other algorithms are statistically significant.

Parameter Sensitivity Analysis:
An analysis of the sensitivity of the algorithm's performance to its parameters should be included. This would provide insights into the robustness of the FA and its dependency on parameter settings, which is vital for practical applications.

Comparison with State-of-the-Art Algorithms:
Ensure that the comparisons are made with a diverse set of state-of-the-art metaheuristic algorithms. This will provide a clearer picture of where FA stands in the current landscape of metaheuristic methods.

Computational Complexity Analysis:
Including a computational complexity analysis of the FA would help in understanding its efficiency in terms of time and space requirements. This is particularly important for its application to large-scale problems.

Experimental design

Please see the basic Reporting

Validity of the findings

Please see the basic Reporting

Additional comments

Please see the basic Reporting

Cite this review as

Reviewer 2 ·

Basic reporting

no comment

Experimental design

no comment

Validity of the findings

no comment

Additional comments

no comment

Cite this review as

---

## Round 0.3 · Minor Revisions

· Academic Editor

Minor Revisions

Dear authors,

Thank you for the revision. It seems that you have not clearly addressed the reviewer’s all comments. You may resubmit the revised manuscript for further consideration following the reviewer’s concerns carefully and submitting a list of responses to the comments along with the revised manuscript. Furthermore, name of the optimization algorithm presented in a new paper “Flood algorithm (FLA): an efficient inspired meta-heuristic for engineering optimization” that is published two weeks ago has the same name with your optimization algorithm. You should mention this state and clearly write the differences for the clear contribution, originality, and overall quality of your paper.

Best wishes,

Reviewer 1 ·

Basic reporting

Overall improved, but not satisfying with mathematical expression and simulation & results. Update references within 2021-2024. If you are unclear about the computational complexity, please review some papers. added other design problems or real-world problems results and comparisons will improve your paper.

Experimental design

add more optimization problems from the literature, apply it with your algorithm, and compare the results from the literature.

Validity of the findings

The comparison was good but need to add more problems with the comparison that will improve your validity of findings.

Cite this review as

---

## Round 0.4 · accepted · Accept

· Academic Editor

Accept

Dear authors,

Thank you for the revision and for clearly addressing all the reviewers' comments. I confirm that the paper is improved. Your paper is now acceptable for publication in light of this revision.

Best wishes,

Reviewer 1 ·

Basic reporting

Updated the paper review

Experimental design

Updated the paper review

Validity of the findings

Updated the paper review

Additional comments

Updated the paper review

Cite this review as